# Knowledge, Attitudes, and Use of Protein Supplements among Saudi Adults: Gender Differences

**DOI:** 10.3390/healthcare10020394

**Published:** 2022-02-19

**Authors:** Manan A. Alhakbany, Hana A. Alzamil, Esraa Alnazzawi, Ghadah Alhenaki, Razan Alzahrani, Aseel Almughaiseeb, Hazzaa M. Al-Hazzaa

**Affiliations:** 1Physiology Department, College of Medicine, King Saud University, Riyadh 11362, Saudi Arabia; malhakbany@ksu.edu.sa (M.A.A.); halzamil@ksu.edu.sa (H.A.A.); 2College of Medicine, King Saud University, Riyadh 11362, Saudi Arabia; esraalnazzawi@gmail.com (E.A.); ghadah.alhenaki@gmail.com (G.A.); razanalz127@gmail.com (R.A.); aseelabdulaziz289@gmail.com (A.A.); 3Lifestyle and Health Research Center, Health Sciences Research Center, Princess Nourah Bint Abdulrahman University, Riyadh 11671, Saudi Arabia

**Keywords:** protein supplements, dietary supplements, gender differences, knowledge, attitude, Saudi Arabia, sports

## Abstract

Saudi Arabia has witnessed recent reforms and positive socio-political changes that have led to increased opportunities for women to participate in fitness centers. This study investigated protein supplement consumption among Saudi females compared with that among males and examined the knowledge and attitudes of the participants toward protein supplement use. A cross-sectional study was conducted in Riyadh using a previously validated, self-administered online survey. The questionnaire included items related to the prevalence, knowledge, attitudes, and practice of protein supplementation. The participants were 354 adults (58.2% were females). The results showed that over 47% of the participants attended fitness centers with more male (80.3%) than female (41%) attendees. Nearly 50% of the participants consumed protein supplements, with males (68.7%) using significantly (*p* < 0.001) more protein supplements than females (35.6%). The powdered form was most commonly consumed. The percentage of gym attendees (67.8%) who used protein supplements was higher than that among non-gym attendees (32.2%). Gaining muscles (56.1%) followed by compensating for protein deficiency (28.6%) were the reasons for taking protein supplements, with a significant gender difference (*p* < 0.001). Coaches provided the most information on protein supplements. The participants appeared to be knowledgeable about protein supplements. Although over 68% of protein supplement users suffered from various symptoms, only 20% of the participants thought that there was no risk in taking protein supplements, with significant gender differences. It was concluded that Saudi male participants are twice more likely to attend the gym and consume protein supplements compared with females. Of both genders, those attending the gym consumed more protein supplements than non-gym attendees.

## 1. Introduction

Dietary supplements (DS) are food products containing dietary ingredients intended to supplement the diet with more nutritional value, and they include vitamins, minerals, amino acids, herbs or botanicals, and other substances [1]. There is a lot of evidence on the widespread use of DS in various countries [2,3,4,5,6,7,8,9]. In Saudi Arabia, many studies have shown that the consumption of DS is becoming increasingly popular, particularly among fitness center attendees. For instance, among a sample of the general public in Riyadh, DS consumption was reported to be prevalent [10]. In addition, half (53.6%) of health science students consumed DS [11]. In a survey from Riyadh, the majority of young adult males used DS, and the most commonly consumed supplements were proteins (29%) [12]. Three other reports conducted among members of fitness centers in Riyadh showed that the prevalence of DS consumption was 37.8% [13], 44.5% [14], and 47.9% [15]. However, other reports from different cities in Saudi Arabia showed diverse proportions. Among health college students in Dammam, the overall prevalence of DS use was reported at 35.91%, whereas intake of whey protein supplement was 23% [16]. In addition, 68.4% of Saudi males attending gyms in Madinah, Saudi Arabia, reported using protein powder supplements [17]. Indeed, Saudi Arabia is the largest market for DS in the Middle East region, and its DS market is expected to reach USD 366.94 million by 2026, growing at an annual rate of 8.04% until 2026 [18].

There are various sources of information about the use of supplements; however, the findings of a study from Lebanon stated that coaches are the major source of information among users [3]. Coaches, as reported by many studies, appear to be the main people recommending the use of protein supplements to the users [19,20]. The majority of coaches and athletic trainers are not specialized in sports nutrition, and they usually have easy access to and financial interest in a wide range of DS and sports performance-related foods, which might have an impact on their consultations [3].

Previous studies have shown that the most commonly consumed supplements among physically active individuals are proteins [21]. According to a recent study, gym visitors prefer whey protein supplements over natural protein sources because they seem to be more effective in increasing post-exercise muscle protein synthesis rates [22]. Whey protein can be rapidly digested and absorbed, and its specific amino acid composition aids in synthesizing muscle proteins [22]. Recent reports indicate that whey protein was the most commonly used dietary supplement among gymnasium users in Riyadh [13,14,17,23].

Earlier studies have also shown that the prevalence of consuming supplements is generally higher in males compared with females [3,4,24,25]. Among US Military Personnel, 42% reported using proteins or amino acids, and factors independently associated with DS use included female gender (OR female/male: 1.91; 95% CI: 1.73, 2.11) [6]. In addition, it appears that there are different reasons for the use of DS among male and female consumers. Males appear to use them as a means of increasing strength, muscle mass, and performance, whereas females use DS for health purposes, recovery from exercise, and to lose weight [3,21,26,27]. In Saudi Arabia, the use of DS is high among males [13,14,17]; however, other studies have shown a high prevalence of DS use among females as well [10,28].

Saudi Arabia has recently witnessed new reforms and positive socio-political changes that have led to more autonomy and opportunities for Saudi women to participate in many societal activities [29]. Physical education programs were introduced in Saudi girls’ schools in 2017, women were permitted to drive in 2018, and licenses were granted to open private fitness centers for women in the same year, thus allowing more prospects for women to engage in sporting activities and to assume an active lifestyle [30,31]. Such increases in women’s participation in fitness centers and health clubs may have led Saudi women to be more exposed to the temptation of protein supplement consumption. The studies that reported limited DS intake by Saudi females were conducted mostly before the recent societal reforms [13,32].

Given the difference in the prevalence of DS consumption between males and females and the increased participation of Saudi females in fitness centers [30,31], the aim of the present study was to investigate protein supplement consumption among Saudi females living in Riyadh compared with that among males and to examine the knowledge and attitudes of the participants toward protein supplement use.

## 2. Methods

### 2.1. Study Design and Participant Selection

This was a cross-sectional study conducted in Riyadh, Saudi Arabia, over a seven-month period from September 2019 to April 2020. Participants were recruited by online distribution using snowball sampling recruitment methods. The recruitment started with the distribution of the online questionnaire to university students through the WhatsApp messaging application. Then, using snowball sampling recruitment methods, the recruitment continued to people in Riyadh other than King Saud University students and staff. We added this information in the revised version of the manuscript.

This study was approved by the institutional review board (IRB) at King Saud University with approval no. E-19-4452. The needed sample size was calculated to be 368 participants on the basis of a 95% confidence level, a population proportion of 0.40, and a margin of error equal to 0.05. Consent to participate in this online questionnaire was built into the questionnaire. The study included males and females living in Riyadh city, and age was reported in two categories: from 18 to 25 years and 26 years and above.

### 2.2. Assessment of Knowledge, Attitudes, and Use of Protein Supplements

A self-administered, pretested, and validated questionnaire was used in the present study to collect information about the knowledge, attitudes, and use of protein supplements by Saudi adult males and females [33]. The questionnaire contained two parts. Part one comprised 19 questions related to general demographic information, including age; sex; living area; frequency of gym visits; protein use, including the amount of protein ingested and required (using specific requirements by grams per kg of body weight); protein type; time and reason for protein consumption; knowledge of participants regarding protein supplement use, such as risks; and benefits and foods containing high amount of protein. Additionally, the source of information, people influencing protein supplement intake, and symptoms associated with protein intake were included in the questionnaire. In part two, the respondents were given eight statements about protein supplements, and they were asked to choose answers using a four-point Likert scale (strongly agree, agree, disagree, and strongly disagree).

### 2.3. Statistical Analysis

Data analyses were conducted using the statistical package for social sciences (SPSS, version 26.0, IBM, Chicago, IL, USA). Descriptive statistics were obtained for all variables and reported as proportions using cross tabulation and Chi-Square tests for the differences relative to gender, protein supplement consumption, and gym attendance. Additionally, logistic regression analysis of selected variables was used relative to gender, adjusting for age as a confounder. The adjusted odds ratios (aORs) and 95% confidence levels are reported. The level of significance was set at a value of ≤0.05.

## 3. Results

A total of 352 respondents agreed to participate in this study (58.2% were females). Table 1 shows the descriptive characteristics of the participants related to gender. Approximately 55% of the sample was 18- to 25-year-olds. Males were significantly (*p* < 0.001) younger than females. Over 47% of the participants attend fitness centers. More males (64.4%) attended the gym than females (34.6%, *p* < 0.001). In addition, males attended the gym more often than females (80.3% versus 41%). Most of those attending fitness centers visited the gym 3–5 times per week (approximately 35% of the total sample). Nearly 32% of the Saudi participants calculated their protein requirements, with significant (*p* < 0.001) differences between males and females. As for protein supplement intake, males (68.7%) had significantly (*p* < 0.001) higher consumption of protein supplements than females (35.6%), with an average intake of 49.4%. In addition, 68.5% of those taking protein supplements suffered from some symptoms due to supplement intake, including diarrhea, constipation, indigestion, nausea, stomach pain, and lack of appetite.

The amount and timing of protein supplement consumption relative to gender are shown in Table 2. The majority (70.1%) of the sample used protein powder, with significant (*p* < 0.001) differences between males and females. Compared with males, females appeared to prefer protein snacks over protein bars. Most of the participants used between one and two scoops of protein supplements per day. Taking protein supplements right after exercise was most common among those using protein supplements. Table 3 presents the cross tabulation of protein supplement consumption with gym attendance among Saudi participants. Compared with non-attendees, both males (22.8% versus 77.2%) and females (45.2% versus 54.8%) who attended the gym consumed significantly (*p* < 0.001) more protein supplements. Overall, approximately 68% of the sample attending the gym consumed protein supplements.

Table 4 displays the participants’ responses to the reasons for taking protein and the benefits and risks of protein supplement consumption by gender. Gaining muscle mass (56.1%) followed by compensating for protein deficiency (28.6%) were the most important reasons for taking the supplements, with significant (*p* < 0.001) gender differences. The findings also showed that 15.1% of the participants were encouraged by their coaches to consume protein supplements. For participants who attended the gym, coaches seemed to provide the participant with the most information on protein supplements. However, nearly half of the sample answered that no one but themselves encouraged them to use protein supplements. In addition, more than one-third of the sample either did not know the risks at all or they knew but did not recognize the exact risks associated with taking protein supplements. Further, 21% of the participant did not think there was a risk at all in taking protein supplements, with more males (30.6%) than females (14.1%) thinking this. Approximately half of the sample believed that gaining muscle mass was the most important benefit associated with using protein supplements.

The responses related to the knowledge and attitudes of participants toward protein supplements among males and females are shown in Table 5. It appears that most participants either disagreed or strongly disagreed on the items that stated that gym attendees should take protein supplements, taking protein supplements minimizes the accumulation of unwanted body fats, and protein supplements are better than protein-rich foods for muscle building. On the other hand, the participants mostly agreed on the items that stated that most people their age do not eat enough dietary protein, protein supplements are necessary for muscle building through weight-lifting, and protein supplements are a good source of energy during workouts.

Finally, Table 6 presents the results of the logistic regression analysis of selected variables relative to gender among Saudi participants, adjusted for participant age. Gym attendance and protein supplements were most significantly associated with male gender. Male participants were more likely to attend the gym (aOR = 2.168; 95% CI = 1.296–3.628, *p* = 0.003) and consume protein supplements (aOR = 2.812; 95% CI = 1.677–4.717), *p* = 0.003) compared with females.

## 4. Discussion

Modern life and increasing dependence on technology has negatively impacted our physical activity and prompted many people to make visiting the gym part of their daily life. Concomitantly, gym attendees and physically active individuals, under the influence of social media, think that they should use dietary supplements, especially proteins and amino acids, to improve their performance and muscle-building. The main findings of the present study indicate that nearly half of the participants use protein supplements with significantly higher protein consumption among males compared with females. Such findings are similar to results reported previously for Saudi participants [10,11,12,13,14,15,16,17]. Previous research in Saudi Arabia showed that the prevalence of protein supplement use increased from 20% to 39% among male gym users [23]. Elsewhere, the use of protein supplements appears widespread among gym attendees [3,4,6,20,21]. Lower consumption of protein supplements than what was shown in our study was reported in a Spanish study, which examined the consumption of protein powder supplement (PPS) among participants of fitness centers and found that 28% of the individuals were using or had used PPS [25]. However, as much as ninety-eight percent of Canadian athletes were reported to consume at least one dietary supplement [34].

The results of the present study confirm that males used protein supplements more often than females. In a previous local study, DS intake was reported to be more common among Saudi males (44.7%) than females (16.4%) who were attending fitness centers in Riyadh [13]. In a study conducted on gym users in a university community in Sharjah, United Arab Emirates, it was found that males (47.7%) consumed more DS than females (28.1%) [32]. Studies in various countries have shown that in general, the prevalence of consuming supplements is higher in males compared with females [3,4,24,25]. In addition, among users of fitness centers in Spain, 42.7% of the total consumers of protein powder supplements were male, and only 3.2% were female [25]. Compared with females, male Canadian athletes were more likely to consume protein powder, energy drinks, branched-chain amino acids, beta-alanine, and glutamine [34]. However, among groups of European endurance runners, sex was not found to be a strong modulator of supplement intake [35].

Furthermore, the majority of our participants, mainly males, reported using one to two scoops of protein powder immediately after exercise, whereas females preferred protein snacks and bars. A previous study found that the majority of gym attendees used protein supplements 1–3 times per week or on a daily basis during the six months prior to enrollment [24]. In our sample, the gender difference could be attributed to the higher gym attendance among males in comparison with females. Additionally, compared with females, the majority of males were more concerned about building muscles (71.4% versus 37% for females) and were more aware of the fact that protein supplements could help them gain muscle mass. On the other hand, females took protein supplements in order to gain strength (9.6% versus 4.3% for males) and improve their shape (8.2% versus 5.5% for males).

Nearly 50% of the participants in our study used the gym three or more days per week. Previous studies conducted among male gym users in Riyadh have shown that the majority visited the gym more than three times per week [14,36]. Additionally, another study reported a significant association between the duration of exercise and the use of hormones and nutritional supplements among male gym members from Riyadh city [15]. In addition, among gym users from three countries (Italy, Turkey, and the UK), it was revealed that gym users who exercised more and consumed higher quantities of protein from foods were more likely to use protein supplements [4]. In another study involving Swiss fitness center clients, it was shown DS intake was correlated with training frequency and that 49% of the participants used protein supplements [20].

In the present investigation, approximately half of the sample believed that gaining muscle mass was the most important benefit associated with using protein supplements. Protein supplement consumption appears to have a positive effect on body composition and muscle strength and performance. Findings from a randomized controlled trial demonstrated positive changes in body composition in young sedentary males and females and strength gains in males after receiving protein supplements during a concurrent training program [37]. Protein supplementation has been reported to have an anti-sarcopenic stimulus and may also aid young and old people taking part in resistance training programs to increase strength and lean body mass [27]. The recommendations from the sports nutrition society indicated that athletes need to have a daily protein intake ranging from 1.4 to 2.0 g protein per kg of body weight [38]. In our study, however, only a third of participants, mainly males, calculated their daily protein requirements.

According to the present findings, gaining muscle mass followed by compensating for protein deficiency were the most important reasons for taking the supplements, with significant differences between males and females. In one study, Canadian athletes indicated that the reasons for DS use were staying healthy, increasing energy, boosting their immune system, enhancing recovery, and improving overall performance [34]. In addition, several studies have reported different reasons for the use of DS among males and females. Males appear to use them as a means of increasing strength, muscle mass, and performance, whereas females use DS for health purposes, recovery from exercise, and to lose weight [26,27]. In the present research, nearly 29% of the participants (more females than males) used protein supplements to compensate for protein deficiency from foods. However, one previous local study involving healthy Saudi adults showed that protein intakes from food (grams per day) were 106.8 ± 4.7 and 92.2 ± 6.1 for males and females, respectively [39]. Protein intake in this study represented, on average, 20.1% of total daily energy intake [39]. This indicates that Saudis, in general, consume enough protein per day from foods.

In the current study, coaches appeared to provide the participants with the most information on protein supplements. The source of information seems to be in line with other published studies that showed that coaches were the principal people recommending the use of protein supplements to users, followed by each person’s individual choice [19,20,32]. Male gym members from Riyadh received their information from non-health professionals [15]. In line with our findings, nearly 30% of the members of Swiss fitness centers obtained their information on DS from the coach or trainer [20]. Among Canadian athletes, the primary sources of information were family and friends, coaches, and athletic trainers [34]. In addition, among gym users from Italy, Turkey, and the UK, coaches were found to be the main source of suggestion for protein supplement intake [4]. Our findings indicate that the participants in this study were fairly knowledgeable about protein supplements. A recent study conducted in Riyadh showed that 88.5% of males and females from the general public had good knowledge of DS [10].

In the current research, 21% of the participants did not think there was any risk at all in taking protein supplements. Yet, a previous study conducted on Saudi gym users showed that the majority of those who consumed protein powder supplements suffered from gastrointestinal symptoms, such as diarrhea, constipation, indigestion, stomach pain, nausea, and decreased appetite. Consuming more than the maximum recommended dose of protein supplements was found to be more common among the Saudi general population compared with medical students [23]. Dehydration is a known side effect of high protein powder consumption [25]. A case study showed that over-the-counter protein supplements interfered with oral L-thyroxine absorption in a middle-aged woman with hypothyroidism and stable thyroid-stimulating hormone [40]. Another study concluded that the prevalence of microalbuminuria in gym attendees was higher than that of the general healthy population but was not associated with protein supplement use [41]. Moreover, a high-protein diet was shown to increase homocysteine concentrations throughout the day but did not increase fasting homocysteine concentrations [42]. However, the clinical relevance of this finding was not investigated in the study, although a high level of homocysteine is considered a cardiovascular risk factor [43]. A systematic review and meta-analysis study showed that dietary protein at levels above the current recommended dietary allowance may be beneficial in preventing hip fractures and bone mass density loss [44].

The present study has both strengths and limitations that should be acknowledged. The strengths include the fact that the study used a validated and pre-tested self-administered questionnaire covering the prevalence, knowledge, and attitudes of the participants toward protein supplement use. The present study also has some notable limitations. The cross-sectional design precludes a causal inference about the temporal sequence of cause and effect. In addition, the convenience online snowball sampling in a single city may limit the generalizability of the findings to all regions of the country, although Riyadh is a cosmopolitan city with people from all parts of the country. Furthermore, we recognize the limitations of self-reported questionnaires.

## 5. Conclusions

The present study showed that male participants are twice as likely to attend the gym and consume protein supplements compared with females. For both genders, those who attended the gym consumed more protein supplements than non-gym attendees. The powdered form was most commonly consumed by participants. Coaches appeared to provide the participants with the most information on protein supplements. Gaining muscle mass followed by compensating for protein deficiency were the most common reasons for taking protein supplements, with a significant gender difference. In addition, the participants appeared to be fairly knowledgeable about protein supplements, and over two-thirds of users suffered from various negative symptoms.

## Figures and Tables

**Table 1 healthcare-10-00394-t001:** Descriptive characteristics of the participants relative to gender (N (%)).

Variable	All (*n* = 352)	Males (*n* = 147)	Females (*n* = 205)	*p*-Value *
Age category				
18–25 years	193 (54.8%)	113 (76.9%)	80 (39.0%)	<0.001
26+ years	159 (45.2%)	34 (23.1%)	125 (61.0%)	
Do you regularly attend the gym?				
Yes	166 (47.2%)	95 (64.6%)	71 (34.6%)	<0.001
No	186 (52.8%)	52 (35.4%)	134 (65.4%)	
Gym visits per week				
None	150 (42.6%)	29 (19.7%)	121 (59.0%)	<0.001
1–2	32 (9.1%)	16 (10.9%)	16 (7.8%)	
3–5	123 (34.9%)	73 (49.7%)	50 (24.4%)	
>5	47 (13.4%)	29 (19.7%)	18 (8.8%)	
Do you calculate your daily protein requirements?				
Yes	112 (31.8%)	68 (46.3%)	44 (21.5%)	<0.001
No	240 (68.2%)	79 (53.7%)	161 (78.5%)	
Do you consume any protein supplements?				
Yes	174 (49.4%)	101 (68.7%)	73 (35.6%)	<0.001
No	178 (50.6%)	46 (31.3%)	132 (64.4%)	

* Chi Squares tests for the proportion between males and females.

**Table 2 healthcare-10-00394-t002:** The amount and timing of protein supplement consumption relative to gender (N (%)).

Variable	All	Males	Females	*p*-Value *
What type of protein supplement do you consume?				
Powder	122 (70.1%)	89 (88.1%)	33 (45.2%)	<0.001
Protein bars	17 (9.8%)	7 (6.9%)	10 (3.7%)	
Other protein snacks	35 (20.1%)	5 (5.0%)	30 (41.1%)	
Number of scoops (grams/day)				
Less than one scoop (<24 g)	22 (18.0%)	9 (10.1%)	13 (39.4%)	<0.001
1 scoop (24 g)	52 (42.6%)	36 (40.5%)	16 (48.5%)	
>1–2 scoops (25–48 g)	44 (36.1%)	40 (44.9%)	4 (12.1%)	
>More than 2 scoops (49 + g)	4 (3.3%)	4 (4.5%)	0 (0.0%)	
Timing of consumption				
Early morning	27 (15.0%)	12 (12.5%)	15 (17.9%)	<0.001
Before exercise	37 (20.6%)	19 (19.8%)	18 (21.4%)	
Immediately after exercise	81 (45.0%)	57 (59.4%)	24 (28.6%)	
Other time	35 (19.4%)	8 (8.3%)	27 (32.1%)	

* Chi-square tests for the proportion between males and females.

**Table 3 healthcare-10-00394-t003:** Cross tabulation of protein supplement consumption with gym attendance among participants (N (%)).

Gym Attendance	Protein Supplement Consumption		*p*-Value *
	Yes	No	
Males			
Yes	78 (77.2%)	17 (37.0%)	<0.001
No	23 (22.8%)	29 (63.0%)	
Females			
Yes	40 (54.8%)	31 (23.5%)	<0.001
No	33 (45.2%)	101 (76.5%)	
All			
Yes	118 (67.8%)	48 (27.0%)	<0.001
No	56 (32.2%)	130 (73.0%)	

* Chi-square tests for the proportion.

**Table 4 healthcare-10-00394-t004:** Participants’ responses to questions about the reason for taking protein and benefits and risks of protein supplement consumption relative to gender (N (%)).

Variable	All	Males	Females	*p*-Value *
For what reason do you take protein supplements?				
Gain muscle mass	92 (56.1%)	65 (71.4%)	27 (37.0%)	<0.001
Compensate for protein deficiency	47 (28.6%)	23 (25.3%)	24 (32.9%)	
Gain muscle strength	7 (4.3%)	0 (0.0%)	7 (9.6%)	
Improve shape	9 (5.5%)	3 (3.3%)	6 (8.2%)	
For other reasons	9 (5.5%)	0 (0.0%)	9 (12.3%)	
Who has encouraged you to take protein supplements?				
Coaches	53 (15.1%)	30 (20.5%)	23 (11.2%)	<0.001
Relatives/friends	39 (11.1%)	24 (16.3%)	15 (7.3%)	
Social media/internet	25 (7.1%)	9 (6.1%)	16 (7.8%)	
Health care provider	11 (3.1%)	0 (0.0%)	11 (5.4%)	
Others	51 (14.5%)	34 (23.1%)	17 (8.3%)	
No one	173 (49.1%)	50 (34.0%)	123 (60.0%)	
Are there risks associated with taking protein supplements?				
Do not know	84 (23.9%)	23 (15.6%)	61 (29.8%)	0.002
No risks	74 (21.0%)	45 (30.6%)	29 (14.1%)	
Yes, but do not know exact risk	115 (32.7%)	44 (29.9%)	71 (34.6%)	
Kidney damage	63 (17.9%)	29 (19.7%)	34 (16.6%)	
Dehydration	5 (1.4%)	2 (1.4%)	3 (1.5%)	
Gout	5 (1.4%)	1 (0.7%)	4 (2.0%)	
Others	6 (1.7%)	3 (2.0%)	3 (1.5%)	
Are there benefits associated with taking protein supplements?				
No	17 (4.8%)	10 (6.8%)	7 (3.4%)	<0.001
Do not know	51 (14.5%)	8 (5.4%)	43 (21.0%)	
Yes, but do not know exact benefits	80 (22.2%)	22 (15.0%)	58 (28.3%)	
Gain muscle mass	169 (48.0%)	90 (61.2%)	79 (38.5%)	
Gain muscle strength	5 (1.4%)	2 (1.4%)	3 (1.5%)	
Enhance performance	7 (2.0%)	2 (1.4%)	5 (2.4%)	
Other	23 (6.5%)	13 (8.8%)	10 (4.9%)	

* Chi-square tests for the proportion between males and females.

**Table 5 healthcare-10-00394-t005:** Knowledge and attitudes of participants about protein supplements relative to gender (N (%)).

Variable	All	Males	Females	*p*-Value *
Should Gym attendees take protein supplements?				
Strongly agree	4 (2.7%)	9 (4.4%)	4 (2.7%)	
Agree	34 (23.1%)	80 (39%)	34 (23.1%)	0.009
Disagree	74 (50.3%)	81 (39.5%)	74 (50.3%)	
Strongly disagree	35 (23.8%)	35 (17.1%)	35 (23.8%)	
Taking protein supplements minimizes the accumulation of unwanted body fats				
Strongly agree	1 (0.7%)	8 (3.9%)	1 (0.7%)	
Agree	21 (14.3%)	54 (26.3%)	21 (14.3%)	<0.001
Disagree	56 (38.1%)	94 (45.9%)	56 (38.1%)	
Strongly disagree	69 (46.9%)	49 (23.9%)	69 (46.9%)	
Most people my age do not eat enough dietary protein				
Strongly agree	27 (18.4%)	24 (11.7%)	27 (18.4%)	
Agree	67 (45.6)	102 (49.8%)	67 (45.6)	<0.001
Disagree	48 (32.7%)	47 (22.9%)	48 (32.7%)	
Strongly disagree	5 (3.4%)	32 (15.6%)	5 (3.4%)	
Protein supplements are necessary for muscle building through weight-lifting				
Strongly agree	14 (9.5%)	14 (6.8%)	14 (9.5%)	
Agree	71 (48.3%)	99 (48.3%)	71 (48.3%)	0.795
Disagree	43 (29.3%)	66 (32.2%)	43 (29.3%)	
Strongly disagree	19 (12.9%)	26 (12.7%)	19 (12.9%)	
Protein supplements are better than protein-rich foods for muscle building				
Strongly agree	1 (0.7%)	1 (0.5%)	1 (0.7%)	
Agree	17 (11.6%)	31 (15.1%)	17 (11.6%)	0.310
Disagree	55 (37.4%)	90 (43.9%)	55 (37.4%)	
Strongly disagree	74 (50.3%)	83 (40.5%)	74 (50.3%)	
Protein supplements are a good source of energy during workouts				
Strongly agree	12 (8.2%)	10 (4.9%)	12 (8.2%)	
Agree	65 (44.2%)	110 (53.7%)	65 (44.2%)	0.203
Disagree	47 (32%)	51 (24.9%)	47 (32%)	
Strongly disagree	23 (15.6%)	34 (16.6%)	23 (15.6%)	

* Chi-square tests for the proportion between males and females.

**Table 6 healthcare-10-00394-t006:** Results of logistic regression analysis for selected variables relative to gender among Saudi participants, adjusted for age.

Variable	Males Versus Females *			
	aOR	(95% CI)	SEE	*p*-Value
Age	0.212	0.128–0.350	0.399	<0.001
Gym attendance (reference = No)	1.00			
Yes	2.168	1.296–3.628	0.263	0.003
Protein supplement consumption (reference = No)	1.00			
Yes	2.812	1.677–4.717	0.264	<0.001
Calculating daily protein requirements (reference = No)	1.00			
Yes	1.620	0.897–2.926	0.302	0.110

* Female was used as a reference category. aOR = adjusted odds ratio; CI = confidence interval; SEE = standard error.

## Data Availability

All data generated or analyzed during this study are included in this published article. Any additional data will be available from the corresponding author upon reasonable request.

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
