# Peer review of "Knowledge, Attitudes, and Use of Protein Supplements among Saudi Adults: Gender Differences"

_healthcare, 2022, doi:10.3390/healthcare10020394_

Round 1

Reviewer 1 Report

The lack of exercise in today's lifestyle is increasingly compensated by the services offered by fitness centers. These exercise facilities also offer a variety of dietary supplements, where protein substrates clearly predominate to increase muscle mass. It is good that fitness centers for women also appear in the Arab world. Unfortunately, even here we encounter problems that occur worldwide, and it is inappropriate or excessive consumption of protein supplements, which often leads to damage mainly to the kidneys of consumers. In my work, I recommend dealing more with sources of information about diet, qualifications and knowledge of instructors in this area. It would also be interesting to know about the interest in these services in the general population, whether the level of education, the availability of centers, the general education associated with the increase in physical activity plays a role. All this will need to be supplemented and analyzed in the discussion, to analyze more social background and especially the possibilities of disseminating quality information among the population. Therefore, I recommend supplementing the results and especially finishing the discussion in the context of the above remarks.

Author Response

Review Report -1

Reviewer’s comment:

The lack of exercise in today's lifestyle is increasingly compensated by the services offered by fitness centers. These exercise facilities also offer a variety of dietary supplements, where protein substrates clearly predominate to increase muscle mass. It is good that fitness centers for women also appear in the Arab world. Unfortunately, even here we encounter problems that occur worldwide, and it is inappropriate or excessive consumption of protein supplements, which often leads to damage mainly to the kidneys of consumers. In my work, I recommend dealing more with sources of information about diet, qualifications and knowledge of instructors in this area. It would also be interesting to know about the interest in these services in the general population, whether the level of education, the availability of centers, the general education associated with the increase in physical activity plays a role. All this will need to be supplemented and analyzed in the discussion, to analyze more social background and especially the possibilities of disseminating quality information among the population. Therefore, I recommend supplementing the results and especially finishing the discussion in the context of the above remarks.

Authors’ Response:

We thank the reviewer for these suggestions and agree that some of the information mentioned in the reviewer’s comments such as those related to diet, fitness instructor’s qualification, participants’ source of information, availability of centers, interest of general population in fitness services and the availability of the services are very important issues. However, in our paper we presented the results and discussed the findings of some of them such as the usage of Gym by the participants, reasons for using protein supplements, sources of information about protein supplements, as well as their knowledge and attitude toward protein supplement use. However, questions such as the availability of and interest in the Gym, the relationship between education and physical activity, and Gym instructors’ education are all beyond the scope of this study. 

Reviewer’s comment:

Does the introduction provide sufficient background and include all relevant references?

Are the conclusions supported by the results?

Authors’ Response:

Yes, we believe that the introduction provided sufficient background about this topic. Also, the conclusions were very much supported by our findings. 

Reviewer 2 Report

Thank you for your interesting survey article.

The basic idea and the design of the survey is very simple and most of the results could be predictable, however, this trial seems to be meaningful for Saudi, which has different social background. Moreover, analysis of the results and the discussions are professional. Here are some minor things to modify.

Line 21. sample ==> people

line 55. chief? ==> major

line 132 ; How did the Gym people calculate their protein requirement?

line 190. wrong position of parentheses

Many mistakes in references. ex) 26, 28, 35 and many more.

Author Response

Review Report -2

Reviewer’s comment:

Here are some minor things to modify.

Line 21. sample ==> people

Response to Reviewer:

We replaced “sample” with “participants”

Reviewer’s comment:

line 55. chief? ==> major       

Response to Reviewer:

We replaced “chief” with “major”, as suggested by the reviewer.

Reviewer’s comment:

line 132 ; How did the Gym people calculate their protein requirement?   

Response to Reviewer:

We added the following sentence that is related to this point in the methods section, under “Assessment of Knowledge, Attitude, and Use of Protein Supplements”: [using specific requirements by grams per kg of body weight].

Reviewer’s comment:

line 190. wrong position of parentheses

Response to Reviewer:

Thanks for the comment. We have corrected the position of the information within the parentheses (lines 193-194).

Reviewer’s comment:

Many mistakes in references. ex) 26, 28, 35 and many more.        

Response to Reviewer:

Thanks for picking up these mistakes.

Reference No. 26: it is correct.

Reference No. 28: we removed the duplicate journal title.

Reference No. 35: In the article title, we replaced capital letters with small letters.

We also checked the rest of the references.

Reviewer 3 Report

Overall an interesting study, there are a number of areas that should be strengthened to make it clearer for readers.

  1. There is work needed to ensure that the language is clear and the meaning unambiguous, for example Line 20 “Nearly 50% of the sample used protein supplements, with males (68.7%) consuming significantly …” it is unclear where this is referring to the amount of supplementary consumed or the number (percent) of each sex who use supplements. Similarly on line 23 “Gym attendants consumed higher protein supplements (67.8%) than non-gym attendants (32.2%)” – does this refer to a higher amount of supplements being consumed or to more people who attend gyms (attendees not attendants) using supplements than those who don’t attend gyms.
  2. Please be clear whether you are talking about sex or gender – they are not the same thing.
  3. Line 43 “In addition, more than half (53.6%) of the health sciences students consumed DS [11].” – I think it is questionable whether you can claim ‘more than half’ when the percent is 54% (n=74).
  4. Line 108 – it is stated that this study is exploring use in young Saudi males and females however the inclusion criteria in line 101/102 states that it is individuals older than 18 and there is no upper age limit.
  5. Please provide more information about participant recruitment e.g. where were invitations to participate posted? What online platform was used for the survey etc. This will impact the resulting participant characteristics.
  6. What is the rationale for categorising the responses for age into above or below 26 years? I recommend standard age brackets are used for analysis
  7. Depending on what the recruitment strategy was the title may not be appropriate as the work may not reflect a general population.

Author Response

Review Report -3

Reviewer’s comment:

There is work needed to ensure that the language is clear and the meaning unambiguous, for example Line 20 “Nearly 50% of the sample used protein supplements, with males (68.7%) consuming significantly …” it is unclear where this is referring to the amount of supplementary consumed or the number (percent) of each sex who use supplements.

Response to Reviewer:

As table 1 indicated, this is referring to the percentage of participants using protein supplements. However, we replaced the word “consumed” with the word “used”.

Reviewer’s comment:

Similarly on line 23 “Gym attendants consumed higher protein supplements (67.8%) than non-gym attendants (32.2%)” – does this refer to a higher amount of supplements being consumed or to more people who attend gyms (attendees not attendants) using supplements than those who don’t attend gyms.       

Response to Reviewer:

We rephrase the sentences to make them clearer. Now, it is written as “The percentage of Gym attendees (67.8%) who used protein supplements was higher than among non-gym attendees (32.2%)”.   

Also, as you can see “attendants” was replaced with “attendees”. 

Reviewer’s comment:

Please be clear whether you are talking about sex or gender – they are not the same thing.       

Response to Reviewer:

We understand that sex refers to person's physical characteristics at birth and include physical and physiological features, while gender includes a person's identities, expressions, behaviors, and societal roles. Findings in the tables’ included mostly information related to attitudes, knowledge and behaviors of the participants.  

Reviewer’s comment:

Line 43 “In addition, more than half (53.6%) of the health sciences students consumed DS [11].” – I think it is questionable whether you can claim ‘more than half’ when the percent is 54% (n=74).    

 Response to Reviewer:

We have re-written this sentence in the revised version of the manuscript as “half”.

Reviewer’s comment:

Line 108 – it is stated that this study is exploring use in young Saudi males and females however the inclusion criteria in line 101/102 states that it is individuals older than 18 and there is no upper age limit.           

Response to Reviewer:

We corrected the sentence in line 107 to “Saudi adult males and females” and removed the word “young”.

We also, added the following sentence: “and age was reported in two categories as from 18 to 25 years or 26 years and above.”  

Reviewer’s comment:

Please provide more information about participant recruitment e.g. where were invitations to participate posted? What online platform was used for the survey etc. This will impact the resulting participant characteristics.          

Response to Reviewer:

Participants were recruited by online distribution using snowball sampling recruitment methods. This statement was added to the revised manuscript, pages 97-98. 

Reviewer’s comment:

What is the rationale for categorising the responses for age into above or below 26 years? I recommend standard age brackets are used for analysis.

Response to Reviewer:

The majority of the participants are young, (below 30 year-olds), so we used this artificial cut-off in order to minimize the large differences that may occur between the numbers of participants below and above this cut-off level.

Reviewer’s comment:

Depending on what the recruitment strategy was the title may not be appropriate as the work may not reflect a general population. 

Response to Reviewer:

We stated in the limitations section that “the convenience online sampling in a single city may limit the generalizability of the findings to all regions of the country, although Riyadh city is a cosmopolitan city with people from all parts of the country.  

Reviewer’s comment:

Moderate English changes required 

Response to Reviewer:

We went again over the entire text and corrected any possible English language mistakes.

Round 2

Reviewer 1 Report

The text was basically not modified, especially in the chapter of the discussion, where the description of the situation clearly prevails without a hint of an explanation of the reasons for the findings. Their interpretation without further data, which determines the decision to visit the fitness center and consume protein supplements during this visit, is very superficial and irrelevant. It follows from the text that the consumption of protein supplements is similar to other, mostly Western countries, but at the same time it is necessary to realize that this applies only to a certain selected group of people with a similar educational and economic background. This is certainly not the case for the majority of the population of Saudi Arabia or for the population of Riyadh as a whole. At a minimum, this must be stated and discussed in the discussion, otherwise the study makes no sense.

Author Response

Response by authors:

The present study intended to examine the differences between Saudi males and females in protein supplements use and their knowledge and attitudes toward protein supplement use, given the recent increase in physical activity of Saudi women and the latest surge in female fitness attendance. In other words, do such increase in Gym use by Saudi women leads to an increase in protein supplements use?

The current finding did not show great increase in supplement use among Saudi females, and the data from the present findings indicated that the gender difference existed due to the higher gym attendance among males in comparison to females. This is previously confirmed by a study conducted among gym users from three countries (Italy, Turkey and UK), as the study revealed that gym users who exercised more and consumed higher quantities of protein from foods were more likely to use protein supplements [4]. In another study involving Swiss fitness center clients it was shown that DS intake correlated with training frequency [20]. 

In addition, compared to females the majority of males were more concerned about building their muscles (71.4% versus 37% for females) and were more aware of the fact that protein supplements can help them gain muscle mass. On the other hand, females were taking protein supplements in order to gain strength (9.6% versus 4.3% for males) and to improve their shapes (8.2% versus 5.5% for males). All the above information are mentioned in the discussion.

As to the other point raised by the respected reviewer tat pertains to the following statement: “This is certainly not the case for the majority of the population of Saudi Arabia or for the population of Riyadh”. We did not generalized our findings to the Saudi population or even those in Riyadh. Our findings applied to Gym users only. Indeed, we have stated in the limitation section tis point.     

Reviewer 3 Report

Please provide detail about the recruitment of participants, while the manuscript states online distribution there is no detail about how exactly participants were recruited (e.g. gym websites, email to members of a particular club etc) and what software was used for this.

Presentation of the age of participants in Table 1 should follow a standard format (i.e. in 10 y brackets)

Author Response

Comments and Suggestions for Authors

Please provide detail about the recruitment of participants, while the manuscript states online distribution there is no detail about how exactly participants were recruited (e.g. gym websites, email to members of a particular club etc) and what software was used for this.

Response by authors:

The recruitment started by distribution of the online questionnaire to the university students through Watts Apps messaging application. Then, using snowball sampling recruitment methods the recruitment continued to those outside the King Saud University’s students and staff to people in Riyadh. We added such information in the revised version of the manuscript.

Comments and Suggestions for Authors

Presentation of the age of participants in Table 1 should follow a standard format (i.e. in 10 y brackets)

Response by authors:     

Age was originally categorized into 3 categories within the questionnaire format. However, since the third category (the older group) was small, we combined it to the second category. Therefore, we end up with only two groups of age categories, above and below 25 years. Thus, we cannot group the participants by 10 years categories.